# Study on Rheological and Mechanical Properties of Rock-Compound-Additive-Modified Asphalt and Its Mixture

**DOI:** 10.3390/ma16103771

**Published:** 2023-05-16

**Authors:** Yanbing He, Jin Yi, Tuo Huang

**Affiliations:** 1Hunan Expressway Group Co., Ltd., Changsha 410008, China; heyanbing_54188@126.com; 2College of Civil Engineering, Central South University of Forestry & Technology, Changsha 410004, China; yjcc1978@163.com; 3National Engineering Laboratory for Highway Maintenance Technology, School of Traffic and Transportation Engineering, Changsha University of Science & Technology, Changsha 410114, China

**Keywords:** modified asphalt, modified asphalt mixture, rock compound additive, rheological properties, mechanical properties

## Abstract

Rutting is one of the most widespread and severe diseases in the service life of asphalt pavement. Enhancing the high-temperature rheological properties of pavement materials is one of the valid measures that can be used to solve rutting disease. In this research, the laboratory tests were carried out to compare the rheological properties of the different asphalts (including neat asphalt (NA), styrene–butadiene–styrene asphalt (SA), polyethylene asphalt (EA), and rock-compound-additive-modified asphalt (RCA)). Then, the mechanical behaviors of different asphalt mixtures were investigated. The results show that the rheological properties of modified asphalt with a 15% rock compound additive performed better compared with the other forms of modified asphalt. The dynamic shear modulus of 15% RCA is significantly higher than the other three asphalt binders, which is 8.2 times, 8.6 times, and 14.3 times that of the NA, SA, and EA at a temperature of 40 °C, respectively. After adding the rock compound additive, the compressive strength, splitting strength, and fatigue life of the asphalt mixtures were significantly enhanced. The results of this research have practical significance for new materials and structures to improve asphalt pavement resistance to rutting.

## 1. Introduction

Rutting is one of the most widespread and severe diseases in the service life of asphalt pavement [1,2,3]. According to statistics, the proportion of road damage caused by rutting disease has reached 40% in the United States and 80% in Japan. Rutting disease control accounts for about 80% of the total maintenance projects in China [4,5]. The generation of rutting will damage the overall structure of the road surface, resulting in a severe decrease in comfort during driving. Due to the seriousness of the rutting problem, the service life of high-grade highways is significantly reduced, and maintenance costs are increased [6]. The main reason for rutting is the accumulation of structural deformation in the pavement caused by the lack of high-temperature stability in the asphalt material, as well as the compression deformation of the mixture under the dual action of the high-temperature environment and the vehicle load [7]. Therefore, it is important to solve the rutting problem.

In response to the rutting problem, researchers continue to explore new solutions to improve the high-temperature stability of the asphalt mixture. At present, the high-modulus asphalt binders (HMAB) and their mixtures (HMAM), proposed by researchers from France, have received more and more attention [8,9]. In 1980, low-grade asphalt binders were used to pave the road in France [10,11]. The layer has excellent load distribution characteristics and high resistance to permanent deformation, allowing the pavement to withstand heavy traffic loads. Afterwards, the specification for using HMAM was formulated, and there are three main ways to prepare a HMAM, including using hard asphalt, using natural asphalt as a modifier, and adding polyolefin-based admixture [12,13,14]. Hard asphalt was added to asphalt mixes, which can effectively enhance the permanent deformation resistance [15,16,17]. The low-grade high-modulus mixture uses hard bitumen with penetrations ranging from 10 to 20 and 15 to 25. The natural asphalts commonly used to prepare high-modulus asphalt include rock asphalt and lake asphalt [18,19,20]. Polyolefins, as thermoplastics, have been widely applied in producing HMAB and HMAM [21,22,23].

However, the properties of HMAB and HMAM prepared by different methods are also different. Wang et al. [24] prepared SBS-modified asphalt and two kinds of HMAB, including using polyolefin and rock asphalt, respectively. The results showed that the rutting resistance of HMAB is better compared with the SBS. On the other hand, the rutting resistance of polyolefin-modified asphalt is better than that of rock-asphalt-modified asphalt. However, the SBS-modified asphalt showed superior fatigue performance than the two high-modulus asphalts. Yan et al. [14] used three methods to prepare HMAM, including rubberized asphalt, natural asphalt and SBS-modified asphalt, and hard asphalt. The modulus and road performance of HMAM have been tested. The results showed that rubber asphalt has the best performance. Lee et al. [25] used hard asphalt to prepare HMAB and compared the property of conventional unmodified asphalt and high-modulus asphalt through laboratory tests. They believed that high-modulus asphalt can be used as a long-life asphalt pavement material. Lu et al. [26] investigated the modification mechanism and fatigue characteristic of high-modulus rock compound additive (RCA)-modified asphalt and its mixture. The results showed that the fatigue performance of RCA-modified asphalt is better than that of neat asphalt. RCA high-modulus-modified asphalt is a road material with simple production process, good storage stability, and excellent durability. However, there is relatively little research on its rheological and mechanical properties, and conclusions on fatigue performance are not unified [27].

In consideration of above situations, the objective of this research is to assess the rheological and mechanical properties of rock-compound-additive-modified asphalt (RCA) and its mixture (RCAM). Firstly, the rheological properties of RCA were studied through temperature sweep test, strain sweep test, and multiple stress creep recovery test. Its performance was compared with the commonly used neat asphalt, polyethylene (PE)-modified asphalt, and styrene–butadiene–styrene (SBS)-modified asphalt. Then, the optimized preparation process was used to prepare asphalt mixtures, and its mechanical behaviors and fatigue properties were studied. The development of this research is of significance for applying high-modulus asphalt.

## 2. Materials and Methodology

### 2.1. Raw Materials

The neat asphalt used in this research is Zhonghai 70# asphalt. The styrene–butadiene–styrene was D1101 linear SBS. The polyethylene-modified asphalt is polyethylene (PE) from a company in Liaoning. The high-modulus modifier is the rock compound additive (RCA). The main component of RCA is natural rock asphalt prepared by adding nano-polymer materials and stabilizing adhesives using composite technology. According to the requirements of the JTG E20 [28], the performance indicators of the neat asphalt were tested, and the test results are shown in Table 1. The technical indices of different modifiers are shown in Table 2, Table 3 and Table 4.

### 2.2. Preparation of Modified Asphalt

In this research, the high-modulus-modified asphalt was prepared using a high-speed shear mixer. Based on previous experiments and literature research, firstly, the modified asphalts were sheared 60 min in high-speed shearing (4000 r/min) conditions under 170 °C. Then, the shearing rate was decreased to 2000 r/min, the shearing temperature was decreased to 150 °C, and the shearing time was decreased to 30 min. In order to investigate the performance of RCA asphalt, the different modified asphalts were prepared. Among them, the contents of SBS and PE were selected to be 3.5%, and 4%, respectively [29,30]. Figure 1 shows the flow chart of the different modified asphalts’ preparations.

### 2.3. Preparation of Modified Asphalt Mixture

Basalt was used as coarse and fine aggregate. The powder used was limestone powder. The properties of the aggregates are shown in Table 5. The grading type was AC-20, which is suitable for the representative grading of the middle surface layers and lower surface layers of the asphalt pavement. Figure 2 shows the gradation curve. The optimum asphalt content can be determined using the Marshall design method. The asphalt mixture mixing process is as follows: Firstly, the preheated aggregate and modifier were mixed for 15 s. Then, the neat asphalt was added for mixing for 90 s, and the mineral powder was added for mixing for 90 s. The total mixing time of the asphalt mixtures was 195 s, and the mixing temperature was 175 °C. The Marshall tester was used for double-sided compaction 75 times, and the compaction temperature was 165 °C. The content of SBS, PE, and RCA was selected to be 3.5%, and 4%, 15%, respectively. The optimum asphalt contents of SBS, PE, and RCA were 4.5%, 4.6%, and 4.8%, respectively.

### 2.4. Asphalt Binder Performance Test

The penetration, softening point, and ductility of the different modified asphalt was conducted based on the following specifications [28]. Additionally, the strain sweep test, temperature sweep test, and multiple stress creep recovery test (MSCR) were carried out to study the different modified asphalts’ rheological properties according to AASHTO T 350 and ASTM D 7405.

In the strain sweep experiment and temperature sweep test, the sample thickness was 1 mm, the temperature range was 40–90 °C, the loading angular frequency was 10 rad/s. MSCR is a method to test the viscoelastic deformation of asphalt binders under different stress levels. For the MSCR test, the temperature was 64 °C, and the mode was stress control mode. Two different stress levels (including 0.1 kPa and 3.2 kPa) were adopted. The creep loading time was 1 s, and then the unloading recovery time was 9 s. The creep recovery rate (*R*) is the ratio of elastic recovery strain to peak strain of asphalt, which can assess the asphalt binder’s elastic recovery capacity. The stress sensitivity index represents the rheological properties of asphalt change with stress change, which is usually expressed by *R_diff_* and *J_nr-diff_.* The specific calculation method is as follows:(1)$$J_{nr}=\frac{\varepsilon_{u}}{\sigma }$$
(2)$$R=\frac{\left(\varepsilon_{p}-\varepsilon_{u}\right)}{\sigma }\times 100\%$$
(3)$$R_{diff}=\left[\left(R_{0.1}-R_{3.2}\right)/R_{0.1}\right]\times 100\%$$
(4)$$J_{nr-diff}=\left[\left(J_{nr3.2}-J_{nr0.1}\right)/J_{nr0.1}\right]\times 100\%$$
where $\varepsilon_{p}$ is the peak strain, $\varepsilon_{u}$ is the unrecovered strain, σ is the stress, *R*_0.1_ and *R*_3.2_ are the creep recovery rates at 0.1 MPa and 3.2 MPa, respectively. *J_nr_*_0.1_ and *J_nr_*_3.2_ are the irrecoverable creep compliances at 0.1 MPa and 3.2 MPa, respectively.

### 2.5. Asphalt Mixture Performance Test

Mechanical properties tests were carried out (including the uniaxial compression test, splitting strength test, interlayer shear and tensile test, rutting test, and fatigue properties of different asphalt mixtures). In order to study the interactions the between adjacent structural layers of asphalt pavement surface, interlayer tensile testing and shear testing of the asphalt mixture were carried out following AASHTO TP-114. The tensile test and shear test both use a cylinder specimen with a size of 2 × 101.6 × 63.5 mm, and the cylinder specimen was covered with adhesive layer of oil between the layers. However, in the interlayer tensile test, the loading method was vertical loading, and the loading rate was 2 mm/min. In the interlayer shear test, a shearing instrument was used for lateral horizontal loading. The loading rate was 10 mm/min.

The uniaxial compression test was conducted following T 0713-2000 and T 0738-2011. The size of the cylinder specimen was 100 × 100 mm. The loading rate was 2 mm/min. The splitting test was conducted following T 0716-2011. The size of the cylinder specimen was 101.6 × 63.5 mm. The loading rate was 50 mm/min. When the splitting test was used to assess the low-temperature cracking performance, a loading rate of 1 mm/min was used and the test temperature was −10 °C.

The rutting test uses a rutting plate and was conducted following T 0716-2011. The size of the specimen was 300 × 300 × 50 mm. In order to investigate the high-temperature characteristics of different asphalt mixtures, the test temperatures used were 60 °C, 64 °C, and 70 °C, and the number of rolling times was 42 times/min. Dynamic stability is the slope of the relatively stable stage of rutting development, which characterizes the speed of rutting development. The calculation of the dynamic stability is shown in Equation (5)
(5)$$DS=\frac{\left(t_{1}-t_{2}\right)\times N}{d_{2}-d_{1}}\times C_{1}\times C_{2}$$
where *DS* is the dynamic stability; *d_1_* and *d_2_* are the amounts of deformation corresponding to times *t*_1_ and *t*_2_; the value of *C*_1_ and *C*_2_ is 1.0; *N* is the rolling speed of the test wheel, which is 42 times/min.

Trabecular specimens were used in the fatigue test. The size of the specimens was 250 × 40 × 40 mm. The test used a stress control mode, and the stress levels were 0.1, 0.2, 0.3, 0.4, and 0.5 MPa, respectively. The test temperature was 15 °C. The loading waveform was a continuous half-sine wave, and the loading frequency was 10 Hz.

## 3. Results and Discussions

### 3.1. Asphalt Binder Basic Performance Results

In order to compare the test results more clearly, the neat asphalt, SBS-modified asphalt, PE-modified asphalt, and RCA-modified asphalt were recoded as NA, SA, EA, and RCA, respectively. The results of the four asphalt penetrations, softening points, and ductility at 5 °C are shown in Figure 3.

As shown in Figure 3, the softening points of the SA, EA, and RCA improved compared with NA. With the content of the rock compound additive increases, the penetration of RCA gradually decreases. The penetration of 15% RCA and 20% RCA is similar and significantly lower compared with the other asphalt. This is because the content of heavy components in the RCA modifier is higher than that of other asphalt. The modifier incorporation leads to a decrease in the proportion of light components in RCA. As a result, the hardness and viscosity of asphalt increase, while the penetration decreases. The softening points of the 10% RCA, 15% RCA, and 20% RCA are about 1.5 times that of the neat asphalt. It can be noted that the RCA modifier is added to the neat asphalt to enhance high-temperature stability. The ductility of RCA decreases with the increase in the rock compound additive. Meanwhile, the ductility of the EA, 10% RCA, 15% RCA, and 20% RCA is lower than that of the NA and SA, which is consistent with the existing research results [31,32]. Under 5 °C conditions, the ductility performance of RCA is lower compared with NA, SA, and EA, which adversely affects its cracking performance.

### 3.2. Rheological Properties Results

#### 3.2.1. Strain Sweep Results

Strain sweep tests were used to determine the linear viscoelasticity range of asphalt materials in this study [33]. The strain sweep results of the different asphalts are shown in Figure 4.

As shown in Figure 4, with the content of the rock compound additive, the variation trends of the dynamic shear modulus are the same. When the strain is lower than 1%, the dynamic shear modulus changes more smoothly. When the strain increases to a certain range, the dynamic shear modulus begins to decrease significantly. When the strain increases, the dynamic shear modulus decreases significantly, indicating that the asphalt material enters the nonlinear viscoelastic range from the linear viscoelasticity range. In order to ensure that the subsequent rheological properties research test is carried out within the range of different asphalt linear viscoelasticity, according to the asphalt strain scanning test results, the strain of dynamic shear rheological test was controlled to 3%. The results are consistent with AASHTO TP 62-03 and NCHRP 9-19 standards.

#### 3.2.2. Temperature Sweep Test Results

In order to grasp the variations in the rheological properties of different asphalt materials with temperature under high-temperature conditions, a temperature sweep test was conducted in a continuous temperature range. The test results are shown in Figure 5.

It can be seen in Figure 5a that as the content of the rock compound additive increases, the dynamic shear modulus increases and the phase angle decreases. The results show that the rock compound additive improves the rheological properties of asphalt. Further, compared with the different modified asphalt, the NA, SA, EA dynamic shear modulus decreases gradually with the rise in temperature. The bigger the dynamic shear modulus, the greater the stiffness and the stronger the deformation resistance. The results indicate that the deformation resistance of NA, SA, EA, and RCA gradually decreases with the rise in the temperature. The dynamic shear modulus of 15% RCA is significantly higher than the other three asphalt binders, which are 8.2 times, 8.6 times, and 14.3 times that of the NA, SA, and EA, respectively, at the initial temperature. The dynamic shear modulus of NA, SA, and EA was 66.51 kPa, 63.25 kPa, and 38.29 kPa at 40 °C, respectively. However, the degree of decrease in the dynamic shear modulus of NA was slightly greater than that of SA, EA, and RCA with the rise in the temperature, especially at 60 °C to 90 °C. This phenomenon may be due to the fact that the polymer cross-linked network inside the SA, EA, and RCA plays a positive role. It can be concluded that the modifier rock compound additive has a tremendous positive effect on improving high-temperature characteristics.

As shown in Figure 5b, the trends of the phase angles change with temperature. The phase angle change in RCA can be divided into two stages: first, there is a certain increase with the rise in temperature, but then the phase angle decreases when a certain temperature is reached. The peak value temperature of the phase angle was 42 °C. This shows that the internal structure network of RCA gradually shows an elastic trend after reaching the peak temperature. The later phase angle decreases to below 45°, indicating that a stable and uniform cross-linked structural network is formed inside the asphalt, which can maintain good elastic properties. The phase angles of NA and EA show an increasing trend with the increasing temperature, indicating that the two kinds of asphalt gradually approached the viscous state. The internal structural network of EA gradually softened with the rise in temperature. The phase angle of SA was maintained within a certain range during the whole temperature change process, indicating a better viscoelastic state. 

The rutting parameter can be used to reflect the ability to resist rutting disease. According to the phase angle and dynamic shear modulus, the variations in the rutting parameter can be calculated. The larger the rutting parameter, the stronger the rutting resistance. As shown in Figure 5c, with the increase in rock compound additive content, the rutting parameter of RCA increases. The rutting parameters of 15% and 20% RCA are similar. Compared with the different types of asphalt, the rutting parameter of the NA, SA, EA, and RCA decreased with the increasing temperature. The difference between the rutting parameters of RCA and that of NA, SA, and EA becomes larger as the temperature rises. The results indicated that RCA’s high-temperature deformation resistance was significantly improved, which is consistent with the existing research results [34].

#### 3.2.3. Multiple Stress Creep Recovery Test Results

The MSCR test results of NA, SA, EA, 10% RCA, 15% RCA, and 20% RCA are shown in Figure 6.

The lower the *J_nr_* value, the stronger the deformation resistance of the asphalt. As shown in Figure 6a, the *J_nr_* values of the same kind of asphalt are different under the stress levels of 0.1 kPa and 3.2 kPa. The value of *J_nr_*_3.2_ is greater than that of *J_nr_*_0.1_, which shows that the irrecoverable creep compliance of asphalt has excellent correlation with loading stress and increases with the rises in stress. The *J_nr_* value of RCA is lower than the other three kinds of asphalt under the stress level of 0.1 kPa or 3.2 kPa. It shows that the addition of PR helps to enhance the stiffness of the asphalt and reduce the viscous deformation of asphalt; additionally, high-modulus asphalt has stronger deformation resistance, which is consistent with the existing research results [35,36].

The greater the *R* value, the stronger the elastic recovery capacity of the asphalt. Figure 6b shows the creep recovery rate results of four asphalts at 64 °C. It can be seen from Figure 6b that *R*_3.2_ of the same asphalt binder is less than *R*_0.1_. This shows that the load stress has a negative effect on the creep recovery rate of asphalt. This is because, as the asphalt phase angle increases at 64 °C, its elasticity decreases, the elastic recovery ability decreases, and the damage and deformation caused by stress to the asphalt increases. Damage and deformation are challenging issues which require repair in a short period of time; otherwise, they become irrecoverably deformed, resulting in a decrease in asphalt creep recovery rate. This is also the reason why rutting and other diseases occur easily with the impact of heavy traffic.

The stress sensitivity coefficient results of four kinds of asphalt are shown in Figure 6c,d. It can be noted that, with the addition of a PR modifier, the *R_diff_* and *J_nr-diff_* values of RCA decrease compared with NA, SA, and EA, showing that the addition of a rock compound additive modifier can enhance deformation resistance and reduce stress sensitivity. This is because as the volume of asphalt expands, the molecular spacing increases, and the interaction force decreases with the rise in temperature. The addition of rock compound additive reduces the air void between asphalt molecules, which improves the high-temperature stability of the asphalt binder.

### 3.3. Mechanical Properties of Asphalt Mixture

Taking into account the basic performance and rheological performance test results of RCA, when the content of rock compound additive was 15%, the modification effect significantly improved. When the content of rock compound additive continued to increase to 20%, the improvement in modification effect is not significant. Therefore, considering the above performance improvement and engineering economy, the content of rock compound additive must be equal to 15%. Further, the modified asphalt mixture was prepared to study its mechanical properties.

#### 3.3.1. Uniaxial Compression Test Results

The static modulus and dynamic modulus of four kinds of asphalt mixtures were compared in this research. Figure 7 shows the uniaxial compression test results (including modulus and strength). When the temperature was 15 °C, the compressive strength values of the neat asphalt mixture (NAM), SBS asphalt mixture (SAM), PE asphalt mixture (EAM), and 15% RCA asphalt mixture (RCAM) were 5.118 MPa, 5.297 MPa, 6.315 MPa, and 6.712 MPa, respectively. The compressive elastic moduli were 2224 MPa, 2589 MPa, 2968 MPa, and 3276 MPa, respectively. When the temperature was 20 °C, compared with the NAM, the compressive strength values of SAM, EAM, and RCAM increased by 6.19%, 34.01%, and 47.73%, respectively, and the compressive elastic modulus of S SAM, EAM, and RCAM increased by 15.48%, 35.20%, and 57.55%, respectively. The above results show that the modifier rock compound additive can enhance the compressive strength and compressive elastic modulus of the NAM. The dynamic modulus of the NAM, SAM, EAM, and RCAM at 15 °C is shown in Figure 7c. It can be concluded from the results that the dynamic modulus of both EAM and RCAM exceed 14,000 MPa. The results indicated that the EAM and RCAM moduli meet the modulus requirements of high-modulus asphalt mixture. Further, the dynamic modulus of the modified asphalt mixture needs to be measured by conducting a four-point bending test. The following experiments further verify whether it has excellent high-temperature resistance to deformation.

#### 3.3.2. Splitting Strength Test Results

Figure 8 shows the results of the splitting test for NAM, SAM, EAM, and RCAM. It can be noted that the splitting strength of the three modified asphalts (SAM, EAM, and RCAM) are higher than that of the NAM. When the temperatures were 15 °C, 20 °C, and 25 °C, the strength values of the RCAM were 1.91 MPa, 1.76 MPa, and 1.54 MPa, respectively. This is due to the rock compound additive modifier particles’ ability to absorb more bitumen, correspondingly reducing the free bitumen in the asphalt mixture. The relative proportion of structural asphalt significantly increased, enhancing the cohesion between the asphalt binder and the mineral aggregate, thus increasing the splitting strength.

#### 3.3.3. Crack Resistance Test Results

In this research, the splitting test was conducted to assess the low-temperature cracking performance of NAM, SAM, EAM, and RCAM. The maximum flexural–tensile failure strength, strain, and modulus can be obtained. The greater the specimen’s strain during failure, the better the low-temperature cracking resistance. Table 6 shows the crack resistance test results. In terms of the failure strength index, the stiffness modulus is larger, indicating that the asphalt mixture is harder and more brittle. The larger the tensile strain, the greater the allowable deformation of the asphalt mixtures under load. It can be concluded that the low-temperature failure strain of the SAM, EAM, and RCAM is smaller compared with NAM. The stiffness modulus of SAM, EAM, and RCAM increases compared with NAM. However, its low-temperature crack resistance can fully meet the requirements.

#### 3.3.4. Interlayer Tensile and Shear Test Results

The connection between different structural layers of asphalt pavement surface was evaluated by conducting interlayer tensile and shear test. The interlayer shear strength and interlayer tensile strength values are shown in Figure 9. From the test results, it can be concluded that the interlayer shear strengths of NAM, SAM, EAM, and RCAM are 1.32 MPa, 1.48 MPa, 1.51 MPa, and 1.59 MPa, respectively. The rock compound additive increased the shear strength of the asphalt mixture, but the increased effect was not obvious. The shear strengths of the SAM and EAM were similar. The interlayer tensile strengths of NAM, SAM, EAM, and RCAM are 0.95 MPa, 1.05 MPa, 1.13 MPa, and 1.16 MPa, respectively. The three modifiers had no obvious effect on the interlaminar tensile strength values of the specimens.

### 3.4. High-Temperature Deformation Resistance

The total deformation and dynamic stability of NAM, SAM, EAM, and RCAM at different temperatures are shown in Figure 10. The test results showed that the maximum deformations of NAM, SAM, EAM, and RCAM at 60 °C were 3.332 mm, 1.753 mm, 1.553 mm, and 1.065 mm, respectively. The maximum deformations at 64 °C were 5.241 mm, 2.794 mm, 2.521 mm, and 2.001 mm, respectively. The maximum deformations at 70 °C were 9.735 mm, 6.339 mm, 5.131 mm, and 2.968 mm, respectively. It can be concluded that the total deformation of the SAM, EAM, and RCAM at 60 °C decreased by 47.38%, 53.19%, and 68.03%, respectively, compared with NAM. At 70 °C, the maximum reduction in total deformation reached 70%. It was found that the incorporation of a rock compound additive modifier can significantly enhance deformation and rutting resistances. The results indicated that the rock compound additive modifier can enhance the rheological properties well. It can be noted, from the relative deformation rate, that the changing trend of the relative deformation rate and the dynamic stability are precisely the opposite. Compared with Figure 10 and Figure 11, the greater the dynamic stability, the lower the relative deformation rate and the better the high temperature characteristics of the asphalt mixture. When the temperature is equal to 70 °C, the relative deformation rate of RCAM is reduced by 69.15% compared with NAM.

### 3.5. Fatigue Test Results

The fatigue test results are shown in Table 7 and Figure 12. It can be found that, compared with the NAM, the fatigue life of the SAM, EAM, and RCAM significantly increases. A phenomenological method was used to further analyze the fatigue test results. The relationship between the fatigue life of the asphalt mixture and the stress level was established according to Formula (6) [37]. The fitting results are shown in Table 8.
(6)$$N_{f}=n{\left(\frac{1}{\sigma }\right)}^{k}\Rightarrow \mathrm{lg}N_{f}=\mathrm{lg}n-k\mathrm{lg}\sigma$$
where *N_f_* is fatigue life, *n*, *k* is the correlation coefficient with material properties, and σ is the stress level.

From the fitting results of the fatigue equation, the rock compound additive modifier significantly improved the properties of the asphalt mixture. The *k* value represents the sensitivity of the fatigue life to the stress level. It was found that the *k* value of RCAM is the smallest, indicating that fatigue life is less sensitive to stress changes.

## 4. Conclusions

This research used a rock compound additive modifier to prepare modified asphalt and its mixtures. The properties of RCA asphalt and neat asphalt, SBS asphalt, and PE asphalt were compared using an asphalt high-temperature rheological test. The four asphalt mixtures’ mechanical properties and high-temperature deformation resistances were further compared and analyzed. The main conclusions of this research are as follows:
Considering the asphalt binder performance improvement and engineering economy, the content of rock compound additive is 15%. The dynamic shear modulus of RCA is significantly higher than the other three asphalt binders. The high-temperature deformation resistances of RCA have been significantly enhanced compared neat asphalt. This effectively improves the anti-rutting ability of the pavement. The compressive strength, splitting strength, and fatigue life of RCAM have been significantly improved compared with NAM. The splitting test results showed the crack resistance of RCAM can fully meet the requirements. Rock-compound-additive-modified asphalt is suitable for regions with a high average temperature.In this study, the rheological and mechanical properties of different modified asphalts and their mixtures were evaluated and compared by conducting laboratory tests. However, microstructure observation tests are needed to further to explain the underlying modification mechanisms of the different modified asphalts.

## Figures and Tables

**Figure 1 materials-16-03771-f001:**
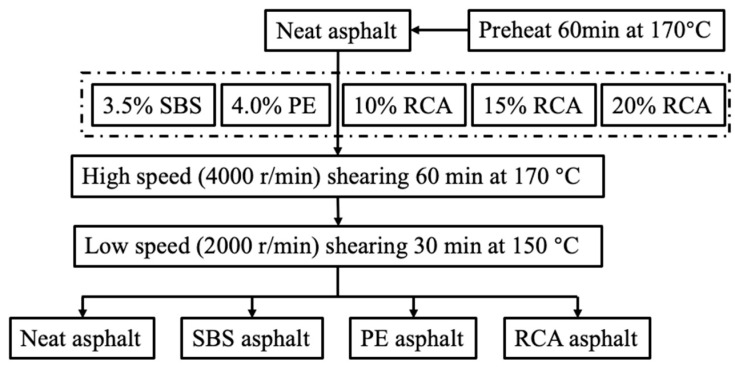
The flow chart of different modified asphalts’ preparations.

**Figure 2 materials-16-03771-f002:**
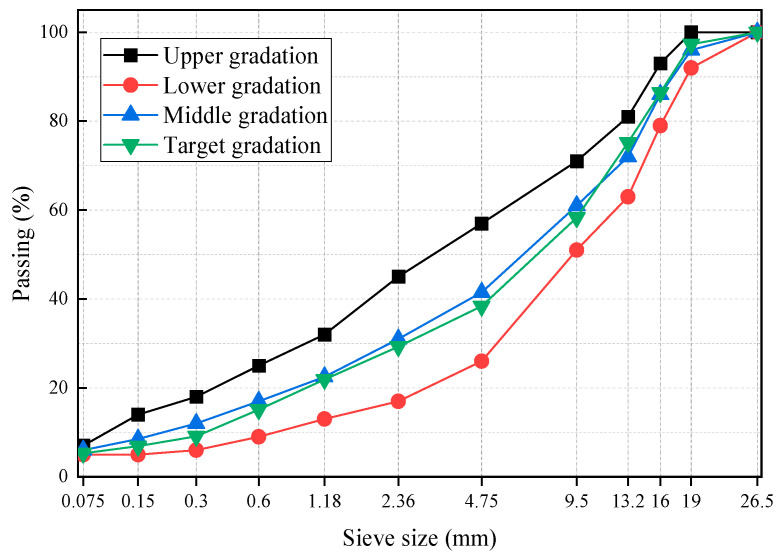
Gradation curve of asphalt mixture.

**Figure 3 materials-16-03771-f003:**
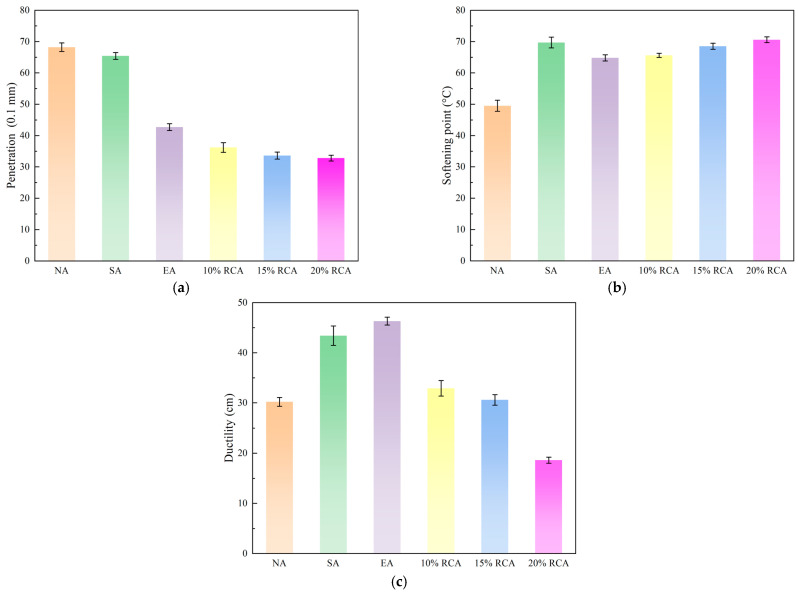
Asphalt binder basic performance test results. (**a**) Penetration, (**b**) softening point, (**c**) ductility at 5 °C.

**Figure 4 materials-16-03771-f004:**
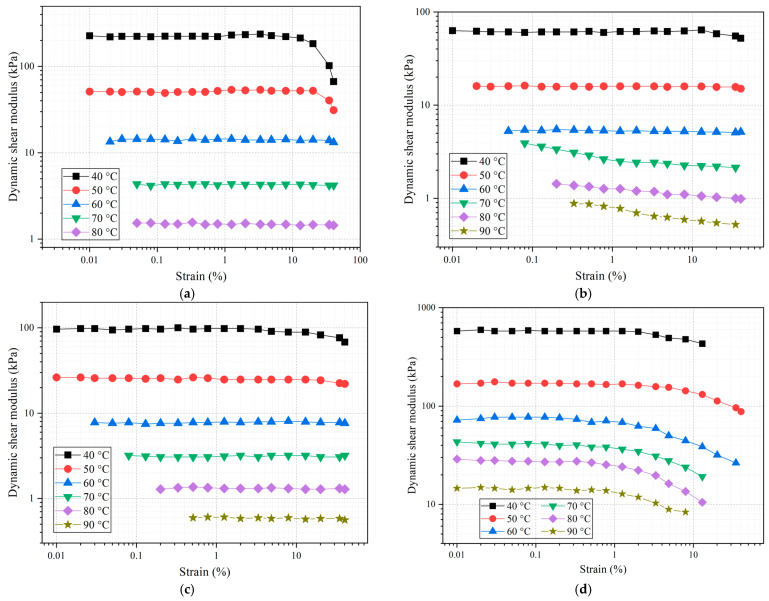
Strain sweep test results. (**a**) NA, (**b**) SA, (**c**) EA, (**d**) 10% RCA, (**e**) 15% RCA, and (**f**) 20% RCA.

**Figure 5 materials-16-03771-f005:**
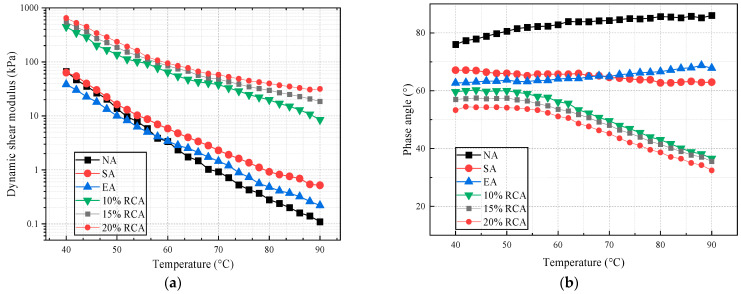
Temperature sweep test results. (**a**) Dynamic shear modulus, (**b**) phase angle, and (**c**) rutting parameter.

**Figure 6 materials-16-03771-f006:**
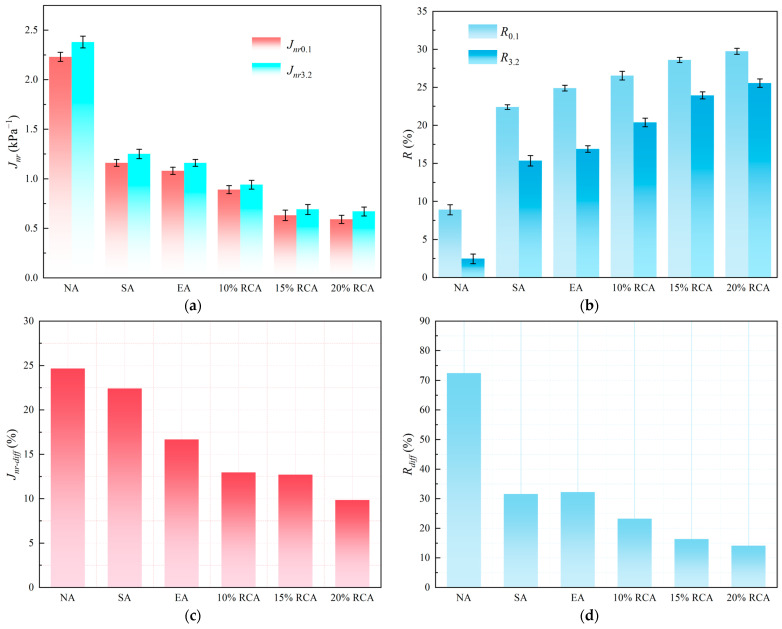
MSCR test results. (**a**) Irrecoverable creep compliance, (**b**) creep recovery rate, (**c**) difference of irrecoverable creep compliance, and (**d**) difference of creep recovery rate.

**Figure 7 materials-16-03771-f007:**
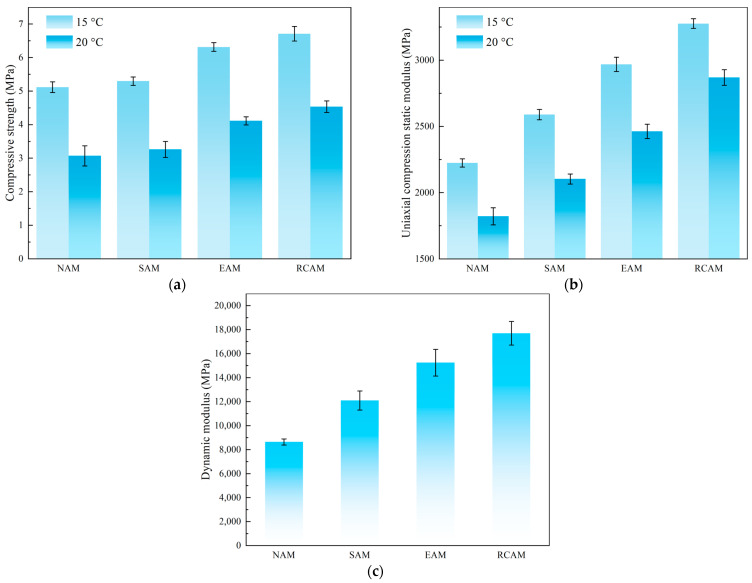
Uniaxial compression test results. (**a**) Uniaxial compression strength, (**b**) uniaxial compression static modulus, and (**c**) uniaxial compression dynamic modulus.

**Figure 8 materials-16-03771-f008:**
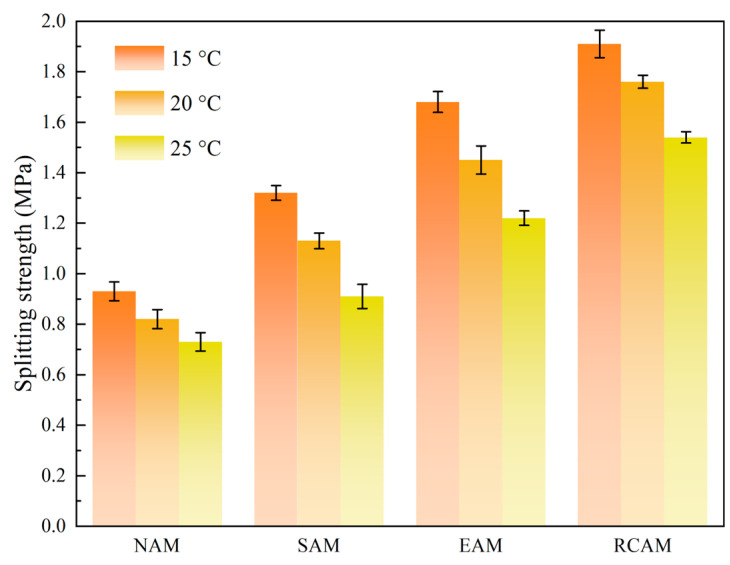
Splitting strength test results.

**Figure 9 materials-16-03771-f009:**
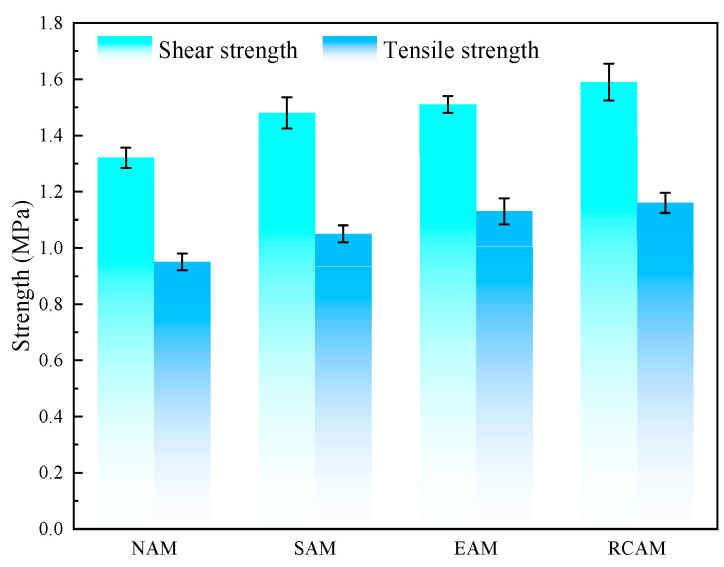
Interlayer tensile and shear test results.

**Figure 10 materials-16-03771-f010:**
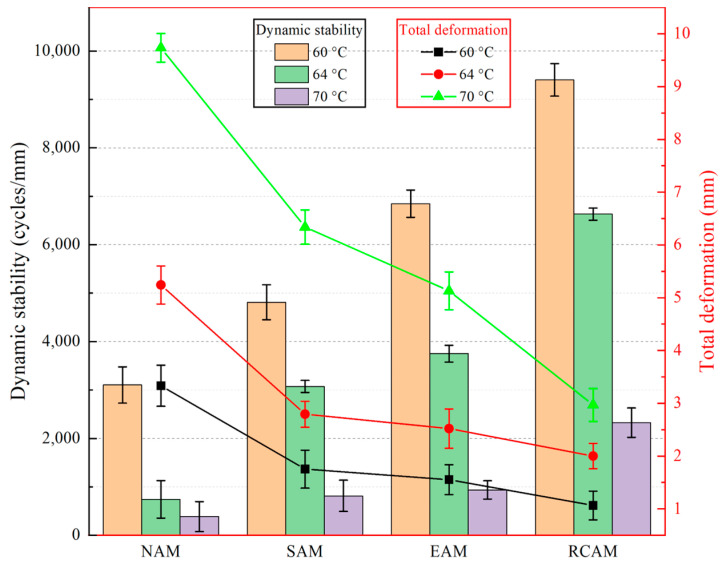
Rutting test results.

**Figure 11 materials-16-03771-f011:**
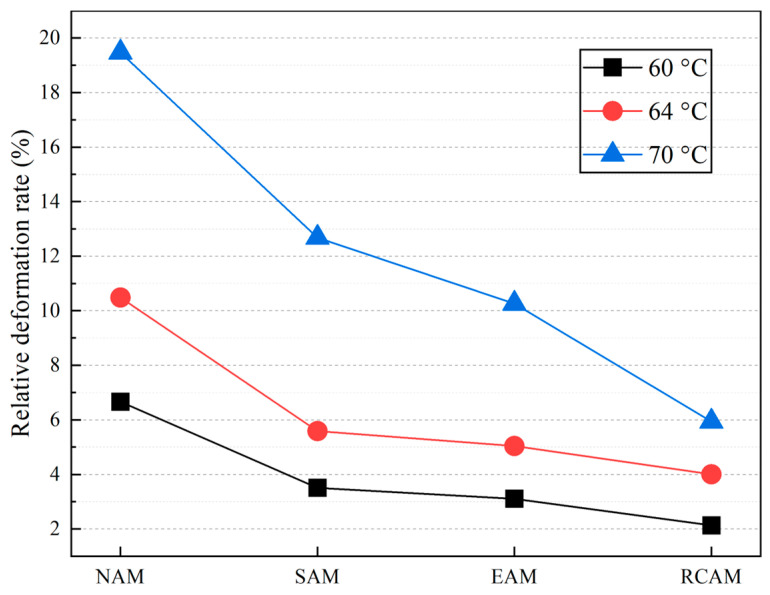
Relative deformation rate.

**Figure 12 materials-16-03771-f012:**
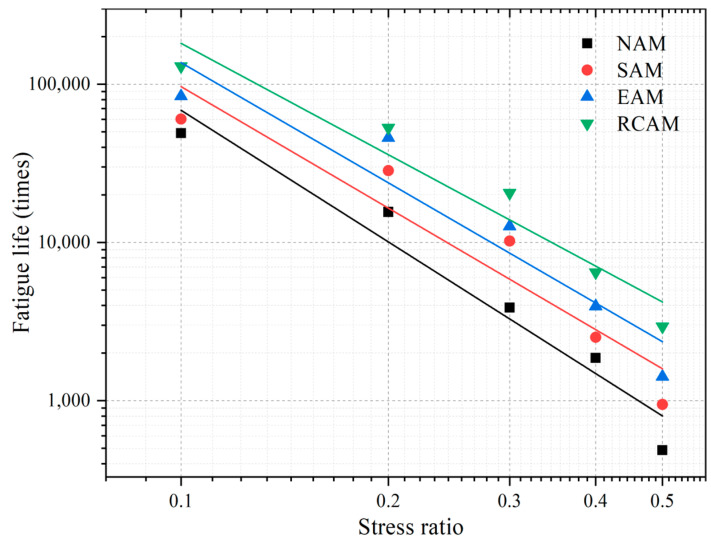
Fatigue test results.

**Table 1 materials-16-03771-t001:** The performance results of neat asphalt.

Test Index		Unit	Test Results	Test Standards
Penetration		0.1 mm	65	T 0604-2011
Penetration Index		-	−0.76	T 0604-2011
Softening point		°C	48.2	T 0606-2011
Ductility	15 °C	cm	134	T 0605-2011
	10 °C	cm	37.6	T 0605-2011
Relative density		-	1.046	T 0603-2011

**Table 2 materials-16-03771-t002:** The technical indices of D1101 linear SBS.

Test Index	Unit	Test Results
Specific gravity	-	0.93
Physical form	-	Powder, pellet
Molecular structure	-	Linear
Tensile strength at break	MPa	31.6
Processing temperature	°C	150–170

**Table 3 materials-16-03771-t003:** The technical indices of polyethylene.

Test Index	Unit	Test Results
Density	g/cm^3^	0.921
Molecular structure	-	Linear
Melt index	g/10 min	18.2
Tensile strength at break	MPa	20
Tensile strength at yield	MPa	10

**Table 4 materials-16-03771-t004:** The technical indices of rock compound additive.

Test Index	Unit	Test Results
Natural rock asphalt content	%	53
Density	g/cm^3^	1.61
Solubility of trichloroethylene	%	54
Flash point	°C	308

**Table 5 materials-16-03771-t005:** The properties of the aggregate.

Test Index	Unit	Test Results	Test Standards
Abrasion value	%	15.2	T 0317-2005
Water absorption	%	0.4	
Crushing value	%	15.7	T 0316-2005
Flat-elongated particles	%	8.2	T 0312-2005
Apparent relative gravity	-	2.709	T 0304-2005

**Table 6 materials-16-03771-t006:** Crack resistance test results.

Types of Asphalt Mixture	Flexural Strength (MPa)	Maximum Flexural Strain (µε)	Stiffness Modulus (MPa)
NAM	5.29	2936	2184
SAM	6.36	2794	2538
EAM	8.24	2683	2842
RCAM	9.78	2574	3469

**Table 7 materials-16-03771-t007:** The fatigue test results.

Asphalt Mixture Type	Stress Level/MPa	Number	Fatigue Life/Times	Average Fatigue Life/Times	Standard Deviation
NAM	0.1	1	53,698	49,034	3691.03
		2	48,732		
		3	44,672		
	0.2	1	18,963	15,595	2387.87
		2	13,698		
		3	14,124		
	0.3	1	4796	3881	723.11
		2	3819		
		3	3028		
	0.4	1	2634	1865	549.56
		2	1578		
		3	1383		
	0.5	1	589	487	79.17
		2	396		
		3	476		
SAM	0.1	1	55,978	60,223	3126.11
		2	63,415		
		3	61,276		
	0.2	1	30,526	28,443	2723.35
		2	24,596		
		3	30,207		
	0.3	1	11,363	10,212	1006.14
		2	8912		
		3	10,361		
	0.4	1	3095	2516	410.74
		2	2186		
		3	2267		
	0.5	1	1285	947	248.87
		2	863		
		3	693		
EAM	0.1	1	94,635	83,838	7799.80
		2	80,395		
		3	76,484		
	0.2	1	49,677	45,790	3443.91
		2	46,388		
		3	41,305		
	0.3	1	11,897	12,649	2457.09
		2	15,963		
		3	10,087		
	0.4	1	3289	3953	629.20
		2	4798		
		3	3772		
	0.5	1	988	1419	307.53
		2	1685		
		3	1584		
RCAM	0.1	1	148,963	129,754	18,594.81
		2	104,596		
		3	135,703		
	0.2	1	61,574	52,971	6160.79
		2	49,863		
		3	47,476		
	0.3	1	24,698	20,551	3203.11
		2	16,899		
		3	20,056		
	0.4	1	5029	6469	1273.62
		2	8126		
		3	6252		
	0.5	1	3498	2940	419.64
		2	2486		
		3	2836		

**Table 8 materials-16-03771-t008:** Fatigue equation fitting results.

Asphalt Mixture	*k*	*n*	*R* ^2^
NAM	2.763	2.071	0.935
SAM	2.551	2.433	0.872
EAM	2.522	2.613	0.873
RCAM	2.338	2.916	0.921

## Data Availability

Data available on request from the authors.

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
