# Peer review of "Study on Rheological and Mechanical Properties of Rock-Compound-Additive-Modified Asphalt and Its Mixture"

_materials, 2023, doi:10.3390/ma16103771_

Round 1

Reviewer 1 Report

Materials

Study on rheological and mechanical properties of rock compound additive modified asphalt and its mixture

Comments:

This paper presents a comprehensive investigation on the rheological performance of rock compound additive modified asphalt compared with fresh, SBS, and PE modified asphalt. The results are systematic and interesting. Some comments are provided for further polishing and improvement.

* Abstract: Please more concise by deleting the detailed information regarding the sample preparation and testing methods. Moreover, the full name of “RCA” in abstract is needed.

* The high-temperature performance improvement of rock compound additive modified asphalt is proved, how about the low-temperature and fatigue properties? It would be better to recommend the application region of this binder.

* Introduction should be improved to show the research gap and objective clearly. Hope these two references can give you guys some improvement ideas. Investigating the effects of SBR on the properties of gilsonite modified asphalt. Construction and Building Materials. Extruded tire crumb-rubber recycled polyethylene melt blend as asphalt composite additive for enhancing the performance of binder. Journal of Materials in Civil Engineering.

* The basic properties of modifiers are required, especially the morphology of rock compound additive.

* Why the RCA dosage is much higher than the SBS and PE? What is the selection basis?

* Why not add some crosslinking agent (like sulfur) during the preparation of SBS modified asphalt?

* How about the compound modification of bitumen with RCA and SBS?

* Some microstructure observation tests are recommended to further explain the underlying modification mechanisms of different modified asphalt.

* The error bar has to be applied in all Figures.

* Table 5: The R2 values of all correlation equations can be added.

* Conclusions: Some recommendations can be provided for future work. In abstract, the optimum dosage of RCA (15%) is mentioned, why not list it as conclusion? How to determine the optimum dosage?

Some language errors have to be revised. 

Reviewer 2 Report

materials-2365156:

*The title should be more specific: what additive?

*Define the acronym RCA in the abstract.

*The abstract is adequate, as well as the keywords.

*The introduction is very generic; should be more focused on the research topic.

In addition, the introduction should make the contribution of the paper clear.

This section should be completely rewritten.

*Present the standards used in the properties of table 1.

*2.2. Preparation of modified asphalt: Justify all parameters (time, temperature, rpm, etc.) adopted in the asphalt modification.

*2.3. Preparation of modified asphalt mixture:

-The authors must present the properties of the aggregates.

-Why was the Marshall method used? It is a disused method. Justify in the text.

*2.4. Asphalt binder performance test:

-Which standard is used for the MSCR? Reference.

*2.5. Asphalt mixture performance test:

-The authors also studied the fatigue of asphalt mixtures; but the introduction only contextualizes permanent deformation. The introduction should address mechanical properties in general (those that were studied in the research).

-Inform the norm for all tests.

*3.1. Asphalt binder basic performance results:

-These are empirical tests; of little importance. They could be excluded from paper.

*3.2.1. Strain sweep results:

-This test should be cited in the Method section.

-The authors use the term "complex modulus"; the correct term is "dynamic shear modulus" for the binder and "dynamic modulus" for the asphalt mix. Correct every document is the figures.

*3.2.2. Temperature sweep test results:

-This test should be cited in the Method section.

-The results should be compared with the existing literature.

*3.2.3. Multiple stress creep recovery test results:

-Present the error (standard deviation) in the graphics; compare the results with the literature.

3.3. Mechanical properties of asphalt mixture:

-Present the error (standard deviation) in the graphics; compare the results with the literature.

-Crack resistance test results: Present the R2 of the fatigue curves (figure 12).

4. Conclusions: Reduce the conclusions. Authors should be more concise.

Moderate editing of English language.

Reviewer 3 Report

The temperature of the fatigue test, the frequency of loading, the method used and the geometry of the sample are decisive for the obtained result. The testing temperature is not found in the article. Studies show that polyethylene is not the best modifier. The main reason is that it does not improve the elasticity of bitumen. Therefore, it would be good to show the percent elastic recovery after MSCR or ductility test.

Asphalt concretes mixtures with RCA bitumen and EA bitumen show better fatigue resistance. If EA does not have pronounced elastic properties compared to SBS, while RCA is the modified one (I assume that it hardens the bitumen), it is not clear how the fatigue resistance improves for these mixtures. Perhaps a higher bitumen content, which is characteristic of HMAC asphalt concretes, can provide it? Thus, typically for HMAC-type asphalt concrete, the stiffness is determined by the 4PB method at 10 degrees Celsius and 10Hz, which should be no lower than 14000MPa. It is not clear whether the modulus determined in this study is comparable to the modulus determined according to 4PB.

It is very desirable to design HMAC-type asphalt concrete with reclaimed asphalt (RAP) and neat bitumen, especially SBS modified. Harder RAP bitumen can provide adequate stiffness, while flexible SBS polymer can help with fatigue resistance. Therefore, it is recommend referring to the DOI: 10.1080/10298436.2020.1850721, where the RAP content in HMAC asphalt concrete is up to 75%, both with unmodified and SBS modified.

Round 2

Reviewer 1 Report

Most comments have been addressed well. 

Author Response

Thank you for your careful review and valuable comments. We will continue to work hard.

Reviewer 2 Report

materials-2365156R1:

The authors carried out an excellent review, finally a last point is requested:

*Reference more studies in this part: “Rutting is one of the most widespread and severe diseases in the service life of asphalt pavement [1].”

https://doi.org/10.1061/(ASCE)MT.1943-5533.0001650

https://doi.org/10.1139/cjce-2015-0546

Minor editing of English language required

Author Response

Thank you for your careful review and valuable comments. 

The References have been added in the revised manuscript. The specific contents are shown as follows.

Rutting is one of the most widespread and severe diseases in the service life of asphalt pavement [1-3]. 

[1] M.Z. Chen, J. Hong, S.P. Wu, W. Lu, G.J. Xu, Optimization of phase change materials used in asphalt pavement to prevent rutting, Advanced materials research 219 (2011) 1375-1378. 10.4028/www.scientific.net/AMR.219-220.1375.

[2] J. Melo, G. Triches, Effects of organophilic nanoclay on the rheological behavior and performance leading to permanent deformation of asphalt mixtures, Journal of Materials in Civil Engineering 28(11) (2016). https://doi.org/10.1061/(ASCE)MT.1943-5533.0001650.

[3] J. Melo, G. Triches, Evaluation of rheological behavior and performance to permanent deformation of nanomodified asphalt mixtures with carbon nanotubes, Canadian Journal of Civil Engineering, 43(5) (2016). https://doi.org/10.1139/cjce-2015-0546.

The manuscript has been checked and revised carefully by native speakers in the USA.

Special thanks to you for your good comments.